# Chip-scale high-peak-power semiconductor/solid-state vertically integrated laser

Jianglin Yue[1], Kenji Tanaka[1], Go Hirano[1], Gen Yonezawa [1], Misaki Shimizu[1], Yasunobu Iwakoshi[1], Hiroshi Tobita[1], Rintaro Koda[2], Yasutaka Higa[2], Hideki Watanabe[2], Katsunori Yanashima[1] & Masanao Kamata [1] ✉

Compact lasers capable of producing kilowatt class peak power are highly desirable for applications in various fields, including laser remote sensing, laser micromachining, and biomedical photonics. In this paper, we propose a high-peak-power chip-scale semiconductor/solid-state vertically integrated laser in which two cavities are optically coupled at the solid-state laser gain medium. The first cavity is for the intra-pumping of ytterbium-doped yttrium aluminum garnet (Yb:YAG) with an electrically driven indium gallium arsenide (InGaAs) quantum well, and the second cavity consists of Yb:YAG and chromium-doped yttrium aluminum garnet (Cr:YAG) for passive Q-switching. The proposed laser produces pulses as short as 450 ps, and an estimated peak power of 57.0 kW with a laser chip dimension of 1 mm$^3$. To the best of our knowledge, this is the first monolithic integration of semiconductor and solid-state laser gain mediums to realize a compact high-peak-power laser.

High-peak-power solid-state lasers have played an important role in modern science and technology, with widespread applications in various fields, including laser remote sensing[1,2], laser micromachining[3,4], and biomedical photonics[5,6]. In line with the progress of semiconductor lasers as pumping sources, the past three decades have seen the emergence of various types of high-peak-power solid-state lasers. These lasers can be classified into categories: laser cavity configurations (fiber lasers[7], thin-disk lasers[8], slab lasers[9], and microchip lasers[10,11], starting from the original rod geometry[12,13]), pulsation methods[14] (Q-switching, cavity-dumping, and mode-locking), etc. Among the pulsation techniques, Q-switching is one of the most successful approaches to obtain kilowatt class laser peak power for science and industry with relative ease, in which a much larger population inversion can be attained by removing cavity feedback to a point, where a sudden change of the cavity Q factor causes the generation of short intense laser pulses. Given that the fluorescent lifetimes of solid-state laser mediums are on the order of microsecond to millisecond[15], which are several magnitudes of orders larger than that of semiconductor gain mediums[16,17], they are suitable for producing short-pulse high-peak-power laser with excellent beam quality. However, unlike semiconductor lasers, which can be electrically driven, the solid-

state laser mediums comprise dielectrics or glasses and require an external laser for pumping, thereby making compact chip-scale high-peak-power lasers very challenging. Therefore, their potential applications are hampered via the system size and assembling cost. Thus, to reach the same maturity level as existing semiconductor lasers, which are suitable for miniaturization and cost-effective mass production, the integration of all elements monolithically compatible with wafer-level manufacturing is required. Meanwhile, as an approach from semiconductor lasers, 2D photonic crystal surface-emitting lasers have recently attracted much attention due to their capability of large-area 2D coherent lasing oscillation of several millimeters, which is much larger than that of conventional semiconductor lasers. A peak power of 18 W has been reported experimentally and up to 300 W can be expected via optimization[18]. Here, we introduce another option for high-peak-power operation exceeding kilowatt, which has not been demonstrated yet for fully integrated chip-scale lasers.

In this work, we originally demonstrate to the best of our knowledge, a fully integrated chip-scale, compact, passively Q-switched laser comprising a semiconductor laser medium and a solid-state laser medium with a peak power, pulse width, and volume of 57.0 kW, 450 ps, and 1 mm$^3$, respectively.

[1]Tokyo Laboratory 04, R&D Center, Sony Group Corporation, Atsugi, Kanagawa 243-0014, Japan. [2]Tokyo Laboratory 06, R&D Center, Sony Group Corporation, Atsugi, Kanagawa 243-0014, Japan. ✉e-mail: masa.kamata@sony.com

# Results

## Concept

To integrate a semiconductor laser and solid-state laser monolithically, a vertical-cavity surface-emitting laser (VCSEL) is more attractive compared to an edge-emitting laser as a pumping laser source because it can emit a laser beam perpendicular to its surface, making it suitable for vertical integration. However, as the volume of the active region of the single-emitter VCSEL is small, the laser output is substantially limited. Therefore, the following problems occur when a solid-state laser is externally pumped. First, it is necessary to use an optical lens to focus the pumping laser beam, which complicates the system configuration and makes vertical integration difficult. Second, as the pumping laser passes through the solid-state laser medium only once or at most twice, the thickness of the solid-state laser medium should be of the same order as the absorption length, which is undesirable from the viewpoint of manufacturing via vertical integration.

To solve these problems, we employ a vertical-external-cavity surface-emitting laser (VECSEL) as an intra-cavity pumping source. Figure 1 illustrates a schematic of our proposed chip-scale semiconductor/solid-state vertically integrated laser, in which two cavities are optically coupled at the solid-state laser gain medium. The first cavity is a VECSEL for intra-pumping of Yb:YAG, with an electrically driven InGaAs quantum well, and the second cavity is for passive Q-switching that comprises Yb:YAG and Cr:YAG. This configuration provides the following advantages. First, as the pumping laser beam is focused via the thermal lens generated at the GaAs substrate and Yb:YAG, the optical lens necessary for external pumping configuration is not required. Second, via intra-pumping, even when the single-pass absorption rate of the Yb:YAG is low, the pumping laser can be efficiently absorbed and the thickness of the Yb:YAG can be reduced. Thus, vertical integration is possible in the proposed configuration.

The VECSEL cavity based on InGaAs quantum well ($\lambda = 940$ nm) comprises two highly reflective (HR) layers at both ends and an intermediate partially reflective (PR) layer. An HR layer is a p-type distributed Bragg reflector (p-DBR), and the other is a dielectric coating layer between the Yb:YAG and Cr:YAG. The intermediate PR layer is an n-type DBR (n-DBR). The VECSEL cavity has no output coupling other than the absorption in the Yb:YAG. The passively Q-switched laser cavity ($\lambda = 1030$ nm) comprises two reflective layers at both ends, where one is an HR layer and the other is a PR layer that acts as an output coupler (OC) for laser emission. The Yb:YAG is selected as a solid-state gain medium, in which high quantum efficiency is expected in the combination of 940 nm pumping and 1030 nm Q-switched laser oscillation. By injecting current into the InGaAs quantum well, which is inside the VECSEL cavity, the 940 nm pumping laser and 1030 nm passively Q-switched laser oscillate successively, thereby causing the emission of a short pulse with high-peak-power.

## VECSEL cavity design

To achieve Q-switched oscillation in the proposed laser configuration, we first design the configuration of the VECSEL cavity. Although there are several advantages with intra-cavity pumping as mentioned above, it has an inherent trade-off. First, since the Yb:YAG is placed in the VECSEL cavity, its absorption for the gain of Q-switched oscillation becomes a loss factor for the pumping laser. Single-emitter VCSEL/VECSEL is quite sensitive to cavity loss due to the small gain; therefore, the introduction of the loss factor should be treated with careful attention. Second, the Yb:YAG obtains the gain of Q-switched oscillation at 1030 nm by absorbing the pumping laser at 940 nm; hence, it is desirable to increase its absorption as much as possible. This balance between the loss of the pumping laser and the gain of the Q-switched laser is one of the important design issues in the proposed laser configuration.

This trade-off can be understood in terms of characteristics of the external cavity part of the VECSEL cavity (Fig. 2a). There are four parameters in the external cavity part: reflectance of the n-DBR ($R_2$), reflectance of the HR layer at 940 nm ($R_3$), absorption rate of the GaAs substrate ($A_{GaAs}$), and absorption rate of the Yb:YAG ($A_{Yb}$). Because $R_3$ should be as high as 100% for effective intra-cavity pumping and $A_{GaAs}$ is specific to the substrate condition, the parameters to be designed are $R_2$ and $A_{Yb}$. However, fully theoretical or numerical analysis of these two parameters is challenging because the dynamic change of the transverse mode at 940 nm must be included due to the thermal lens produced at the GaAs substrate and Yb:YAG, whose simulation is not easy to perform accurately. Instead, to obtain a design guideline for the VECSEL cavity, we develop a semiempirical model that combines experimental data and external cavity theory.

We first discuss the n-DBR reflectance $R_2$. For efficient power build-up of the 940 nm laser in the cavity, $R_2$ should be as high as possible to overcome the loss factor in the external cavity part. However, an excessively high $R_2$ incurs a risk of VCSEL oscillations between the p-DBR and n-DBR (rather than VECSEL oscillations). In this oscillation mode, the pumping of the Yb:YAG is not intra-cavity pumping but external pumping, which prevents the effective pumping-power absorption of the Yb:YAG. Figure 2b shows the measured VECSEL output power for two semiconductor devices with $R_2 = 90.0\%$ and 96.1%. Note that a Yb:YAG was not inserted, and an external OC at 940 nm (reflectance: $R_{940}$) was placed for the VECSEL output oscillation. Between the two n-DBR conditions, a higher output power is obtained with $R_2 = 96.1\%$ as expected. It can also be seen that the output power varies with $R_{940}$. This variation comes from the change in effective reflectivity of the external cavity part ($R_{eff}$) due to $R_{940}$, indicating the proper occurrence of VECSEL oscillations.

We next discuss the Yb:YAG absorption rate $A_{Yb}$ from the viewpoint of the pumping-power absorption of the Yb:YAG ($P_{Yb}$). As explained above, $A_{Yb}$ is directly related to the trade-off between loss of the pumping laser and gain of the Q-switched laser. Figure 2c illustrates the relationship between $A_{Yb}$ and $P_{Yb}$ for $R_2 = 96.1\%$, which was analyzed using the measured output power (Fig. 2b) and the derived equations in the external cavity part (see Methods for details). When $A_{Yb} < 30\%$, $P_{Yb}$ increases with increasing $A_{Yb}$, indicating that a higher Yb:YAG gain is expected at higher $A_{Yb}$. In contrast, when $A_{Yb} > 30\%$, the $P_{Yb}$ decreases with increasing $A_{Yb}$ because $R_{eff}$ becomes smaller with higher $A_{Yb}$; consequently, the condition shifts from the optimum $R_{eff}$ condition.

To confirm the basic gain characteristic of our laser configuration at 1030 nm, a fundamental experiment of 1030 nm continuous wave

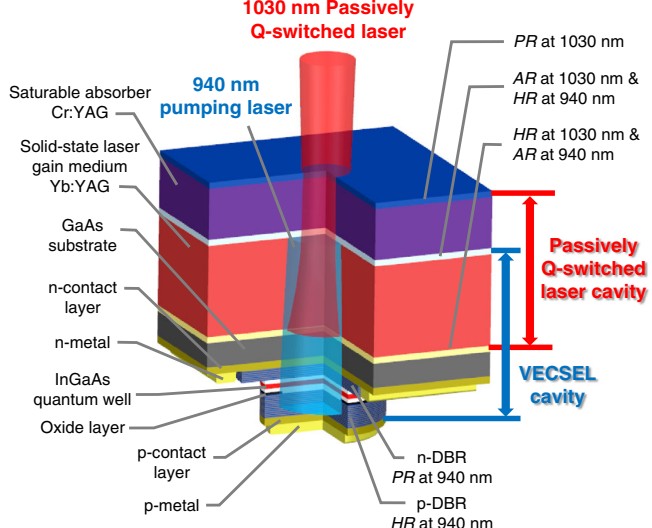

**1030 nm Passively Q-switched laser**

**940 nm pumping laser**

Saturable absorber
Cr:YAG

Solid-state laser gain medium
Yb:YAG

GaAs substrate

n-contact layer

n-metal

InGaAs quantum well

Oxide layer

p-contact layer

p-metal

PR at 1030 nm

AR at 1030 nm & HR at 940 nm

HR at 1030 nm & AR at 940 nm

Passively Q-switched laser cavity

VECSEL cavity

n-DBR
PR at 940 nm

p-DBR
HR at 940 nm

**Fig. 1 | Schematic of chip-scale semiconductor/solid-state vertically integrated laser.** The VECSEL cavity and the passively Q-switched laser cavity are optically coupled at the solid-state gain medium of Yb:YAG.

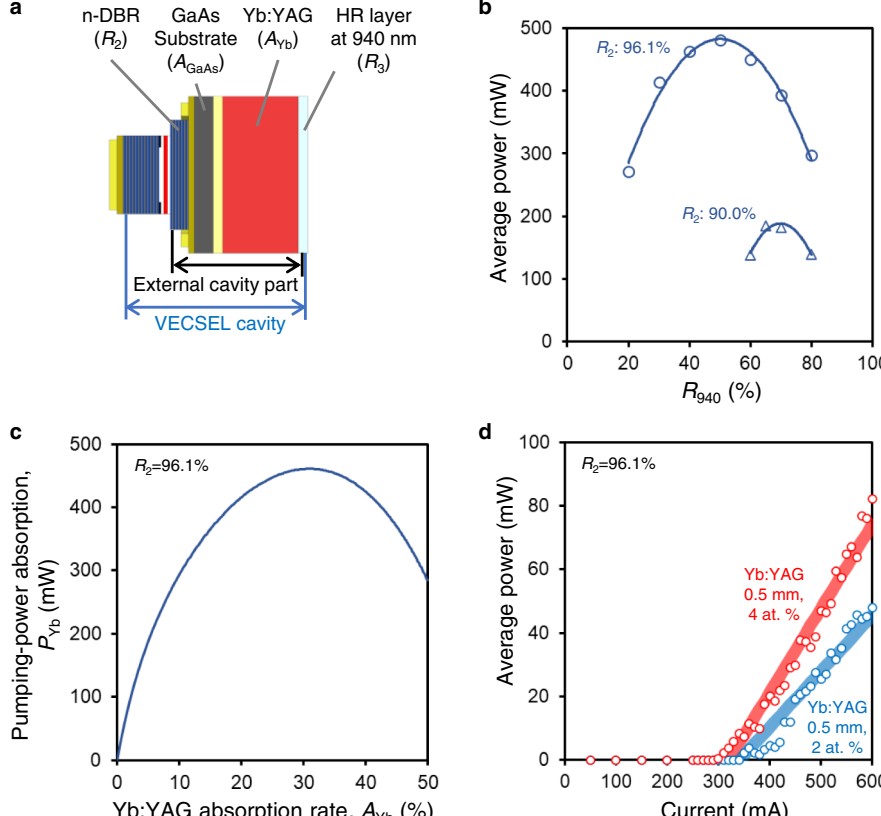

**Fig. 2 | Design of the integrated VECSEL cavity. a** VECSEL cavity model.
**b** Measured VECSEL output power for $R_2$ = 90.0% and 96.1%. The Yb:YAG was
replaced by a 940 nm OC (reflectance: $R_{940}$). The distance between the GaAs sub-
strate and OC was 0.5 mm. **c** Analyzed pumping-power absorption of Yb:YAG ($P_{Yb}$)
versus Yb:YAG absorption rate ($A_{Yb}$) for $R_2$ = 96.1%. Other parameters were set as

follows: $R_3$ = 99.0% and $A_{GaAs}$ = 7.0%. **d** Measured I–L characteristics in the 1030 nm
CW laser oscillation experiment with a 1030 nm OC ($R_{oc}$ = 85%). The Yb:YAG was
0.5 mm thick and its doping concentration was 2 at. % ($A_{Yb}$ = 9.5%) or 4 at. %
($A_{Yb}$ = 18.1%).

(CW) oscillation was performed, in which a 1030 nm laser cavity only
included a Yb:YAG (i.e., no saturable absorber). The detailed condi-
tions were as follows: the thickness of the Yb:YAG was 0.5 mm, and the
doping concentrations were 2 and 4 at. %, which corresponded to $A_{Yb}$
of 9.5% and 18.1%, respectively. Both surfaces of the Yb:YAG were
coated to form the VECSEL and 1030 nm cavities (see Methods for
details of the Yb:YAG coating). An OC with a reflectance ($R_{oc}$) of 85% at
1030 nm was placed additionally. The semiconductor element was
cooled to 20 °C by an active cooling system. Because of the difficulty of
mechanical assembly, the Yb:YAG was placed at a distance of 0.4 mm
from the semiconductor element. Figure 2d shows the current–output
(I–L) characteristics of 1030 nm CW oscillation. A laser oscillation was
confirmed at both doping concentrations. Moreover, a higher output
power was achieved at 4 at. % ($A_{Yb}$ = 18.1%). This result agrees well with
the analyzed pumping-power absorption in Fig. 2c, in which $A_{Yb}$ of
18.1% realizes higher pumping-power absorption than $A_{Yb}$ of 9.5%.

The development of the semiempirical model led to a method for
deriving the VECSEL cavity parameters, especially the two important
parameters $R_2$ and $A_{Yb}$.

**Passively Q-switched laser simulation**
To derive the passively Q-switched laser cavity configuration, we
developed a simultaneous rate equations model between the VECSEL
cavity and passively Q-switched laser cavity. This model can help us
understand the carrier and photon dynamics of both the VECSEL and
passively Q-switched cavities, which are driven electrically and opti-
cally, respectively. More details on this model can be found in the
"Methods" section. Figure 3 illustrates the simulation results of the
model. The configuration of the passively Q-switched laser simulation

consists of 0.5 mm thick Yb:YAG with a doping concentration of 4 at. %,
0.2 mm thick Cr:YAG with initial transmittance of 95%, and the OC with
a reflectance of 85%.

The top part of Fig. 3a shows that when current is injected into the
InGaAs quantum well of the VECSEL at an initial time ($t$ = 0 μs), the
carrier density ($N$) and photon density ($S$) increase and the oscillation
of 940 nm pumping laser begins within a few nanoseconds. Then, the
940 nm pumping laser maintains a stable oscillation state
($t$ = 0–900 μs). The bottom part of Fig. 3a shows that the population
inversion density ($N_g$) increases when the 940 nm pumping laser is
partially absorbed by the Yb:YAG, which means that energy is accu-
mulated in the 1030 nm cavity for several hundred microseconds
($t$ = 0–470 μs). When Cr:YAG is bleached, the cavity Q factor switches
to a high value in a short time. Furthermore, the population inversion
density decreases rapidly, and a high-peak-power laser pulse is gen-
erated with a pulse width of several hundred picoseconds (around
$t$ = 470 μs), as shown in Fig. 3b. After Q-switched laser pulse oscillation,
the saturable absorber becomes unbleached again, the cavity Q factor
switches to a low value and energy is accumulated for the next
Q-switched laser pulse oscillation ($t$ = 470–740 μs), as shown in Fig. 3a.
From the top part of Fig. 3a, it is also found that photon density at
940 nm decreases slightly when 1030 nm picosecond pulses are gen-
erated. This phenomenon occurs because the ground-state population
density of $Yb^{3+}$ ion and the absorption at 940 nm increase by
Q-switched laser pulse oscillation, and the photon density at 940 nm
decreases after laser pulse oscillation.

The simulation results show that in our proposed laser configura-
tion, the 940 nm pumping laser maintains a stable oscillation state, and
1030 nm passively Q-switched laser oscillation is theoretically possible.

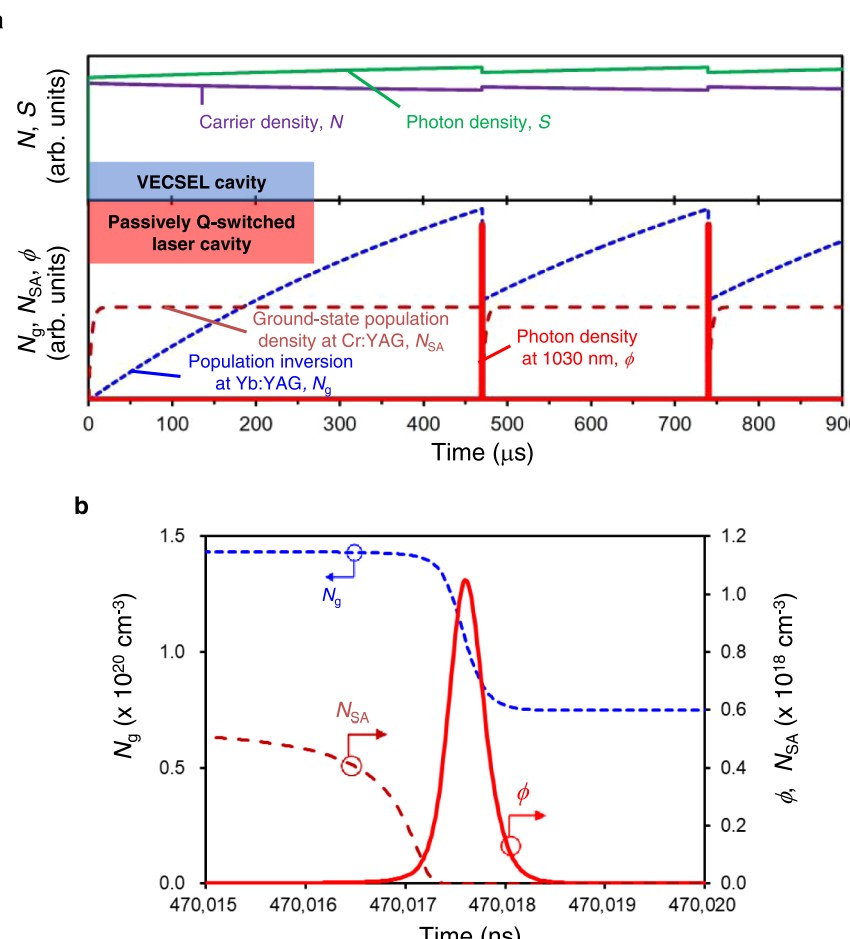

**Fig. 3 | Simulation results of the simultaneous rate equations model. a** (top) Temporal dynamics of the carrier and photon densities at 940 nm in the VECSEL cavity; (bottom) temporal dynamics of the population inversion density at Yb:YAG, ground-state population density at Cr:YAG, and photon density at 1030 nm in the passively Q-switched laser cavity. **b** Temporal dynamics of the population inversion density in Yb:YAG, ground-state population density in Cr:YAG, and the photon density at 1030 nm at the time of Q-switching (~470 μs in **a**).

## Passively Q-switched laser demonstration

To demonstrate the concept of the proposed laser configuration, first, we fabricated a mechanically assembled semiconductor/solid-state vertically integrated laser for the ease of the experiment. Note that all elements were assembled mechanically, and there was a 0.4 mm air gap between the semiconductor element and the Yb:YAG due to the difficulty of the mechanical assembly as already mentioned. The laser device parameters and the cooling condition of the semiconductor element were the same as those described in the previous sections. Figure 4a shows the I–L characteristics, and a laser oscillation is confirmed at a current threshold value of 300 mA. Figure 4b, c show the single-pulse waveform and the temporal waveform, respectively, at an injection current of 370 mA. The temporal pulse profile was measured by a high-speed biased fiber-optic detector (Newport Corporation 818-BB-35F) with a bandwidth of 15 GHz and a digital phosphor oscilloscope (Tektronix TDS7404) with a bandwidth of 4 GHz (Sampling Rate: 20 GHz). The pulse width of 461 ps and the pulse trains (1.2 kHz repetition frequency) are confirmed, indicating that Q-switched oscillation has successfully occurred. The peak power (23.2 kW; see Fig. 4b) was estimated from its temporal pulse waveform with a pulse energy of 12.1 μJ measured by a pyroelectric energy meter (Ophir PE9-C). Figure 4d exhibits the laser spectrum measured under the same current condition. A single longitudinal mode with a peak wavelength at 1030.1 nm (measured by an optical spectrum analyzer with a resolution of 0.1 nm) is confirmed. These measured results demonstrate that

Q-switched oscillation is practically possible in our proposed laser configuration.

Next, we demonstrate Q-switched laser oscillation of a fully integrated 1 mm³ chip-scale laser. The cavity configuration of a fully integrated chip-scale laser differs from that of a mechanically assembled laser (See Methods for details of the device structure). Figure 5a shows the fabricated laser. All elements were successfully bonded together by chip-scale integration, and the laser was mounted on a transistor outline can (TO-can) with a diameter of 9 mm, which is commonly used in semiconductor laser packaging. Figure 5b shows the I–L characteristics after active cooling at 20 °C. A laser oscillation is confirmed at a current threshold of 280 mA. The following laser characteristics were measured under a current injection of 330 mA. The pulse energy remained stable for 2 h (Fig. 5c). The average pulse energy was 30.4 μJ with a standard deviation of 0.2 μJ and the repetition frequency was 1.79 kHz. Figure 5d shows the single-pulse waveform with a pulse width of 450 ps. The estimated peak power was as high as 57.0 kW. A single longitudinal mode at a wavelength of 1030.6 nm and $M^2$ of 1.36 (x axis) and 1.22 (y axis) are also confirmed (Fig. 5e, f). The output power was stable for 24 hours, and passively Q-switched oscillations with an average power of 53.5 mW and standard deviation of 0.3 mW were obtained (Fig. 5g) under the current injection of 330 mA, which is the same condition as the series of the laser characteristics measurements described above. Stability and lifetime tests for different driving currents are also planned for future studies. The series of experimental

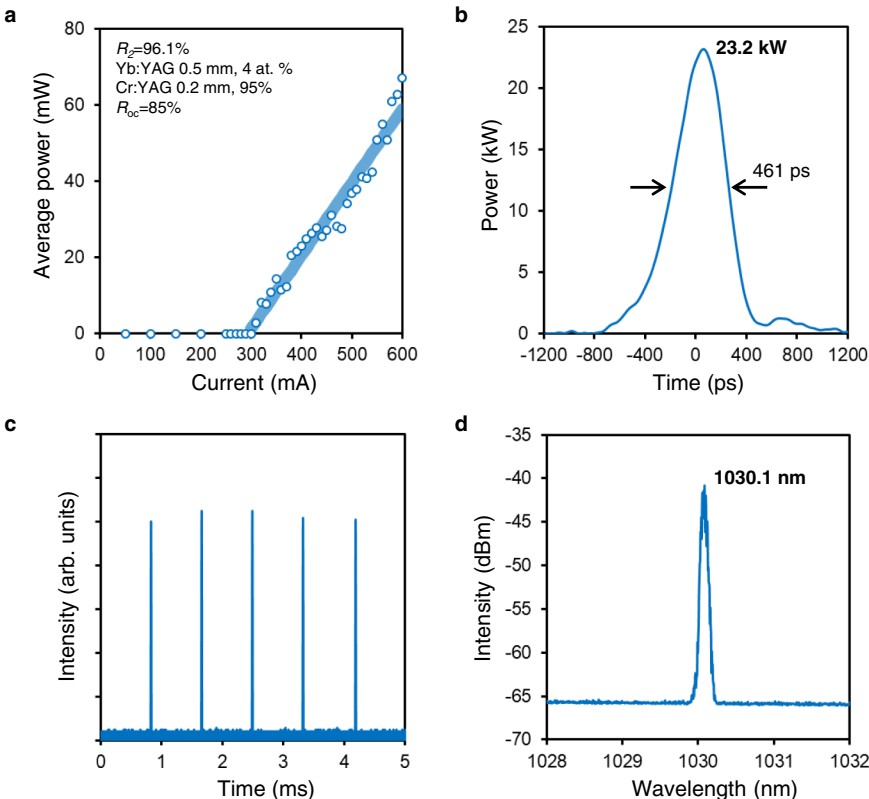

**Fig. 4 | Experimental results of the mechanically assembled laser. a** I−L characteristics. **b** Single-pulse waveform. The vertical axis was calculated from the obtained pulse energy (12.1 μJ) and the pulse waveform. **c** Temporal waveform.

**d** Optical spectrum. The data presented in **b**−**d** were measured at an injection current of 370 mA.

results demonstrate that our proposed laser configuration can realize a high-peak-power over kilowatts at the same chip-scale size as semiconductor lasers.

Some differences exist between the result of the mechanically assembled laser and that of the fully integrated chip-scale laser. In addition to the cavity configuration, these differences might arise from the VECSEL cavity length (air gap), the thermal diffusion effect between the semiconductor element and Yb:YAG, and individual differences among the elements (VECSEL, Yb:YAG, and Cr:YAG). Interaction among these factors should affect the coupled laser oscillation parameters such as pumping laser power, mode matching, absorption cross section, and stimulated emission cross-section of the Yb:YAG and Cr:YAG.

## Discussion

The presented results provide clear evidence that the configuration of chip-scale high-peak-power lasers by vertical integration of semiconductor and solid-state laser gain mediums is possible, thereby bridging the gap between semiconductor lasers and solid-state lasers for the first time since their inventions. Although we chose the combination of InGaAs quantum well and Yb:YAG as gain mediums, other materials can easily be used to generate different laser wavelengths other than 1030 nm, so our approach opens a new door for establishing a versatile platform for compact high-peak-power lasers. By properly taking into consideration injection current distribution and mode matching between the VECSEL cavity and the passive Q-switched laser cavity, further power improvement should be possible by increasing the chip size. Moreover, as the laser is capable of wafer-level vertical integration, it can potentially be coupled with flat optics such as nanophotonic metasurfaces[19] to customize its phase, amplitude, polarization, and beam direction.

This laser can be applied to several applications. Compact high-peak-power lasers are very promising in the field of light detection and ranging (LiDAR) applications for autonomous driving cars, drones, and robots. The use of such lasers in LiDAR applications is necessary to meet the potential demands of a high-peak-power laser source on the size scale of semiconductor lasers, but no laser source currently meets this market demand.

Additionally, the proposed laser differs greatly from conventional solid-state lasers in terms of the ease of array arrangement. The minimum lateral size of our laser is determined via the oxide aperture diameter of the VECSEL, which is about 150 μm. This means that many lasers can be arranged in parallel within one chip. Moreover, since the manufacturing process of the VECSEL uses a photolithography process, the array arrangement can be achieved with high positional accuracy (on the order of micrometers). This laser array device can benefit not only LiDAR, but also laser micromachining used in manufacturing and laser medical applications. In laser micromachining, irradiating a target area with a single laser beam causes significant thermal accumulation at a laser repetition rate of approximately 1 MHz[20]. Therefore, the repetition rate cannot be simply raised to increase the throughput of laser micromachining. In such cases, parallel processing using arrayed laser beams is an effective alternative. Although a beam splitting device for this purpose has been reported[21], it lacks flexibility because each arrayed laser beam cannot be switched independently since only one primary laser light source is used.

We believe that our lasers will transform the situation of conventional high-peak-power lasers, where potential effectiveness has been confirmed at the laboratory level, but actual field applications have been limited by cost and size constraints.

## Methods
### Device structure
Figure 1 illustrates a schematic of the proposed laser. The proposed laser consists of a VECSEL, solid-state laser gain medium, and saturable

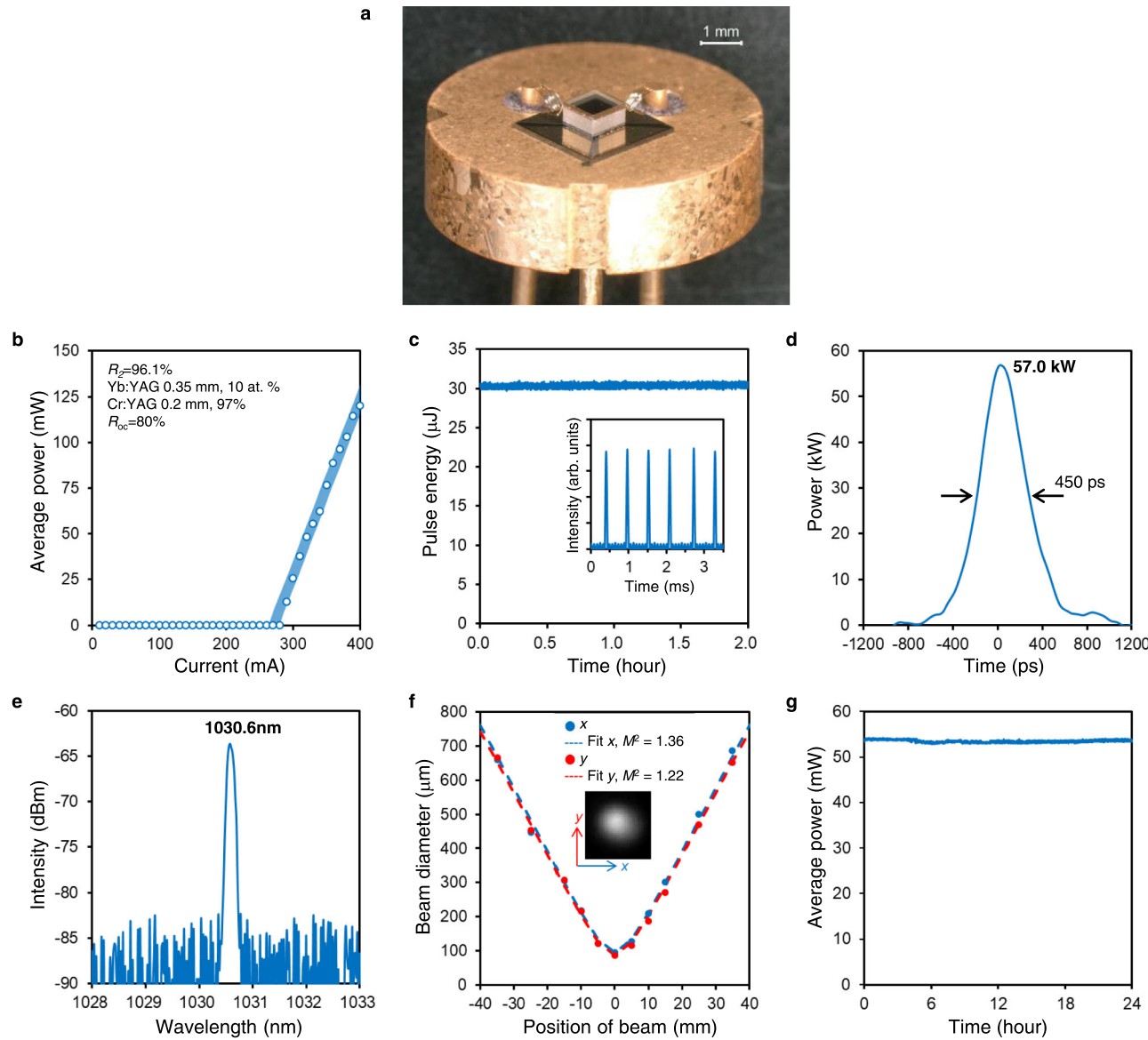

**Fig. 5 | Demonstration of the chip-scale semiconductor/solid-state vertically integrated laser. a** Photograph of the laser chip. **b** I–L characteristics. **c** Pulse energy stability (2 h) of the chip-scale integrated laser (inset: temporal waveform). **d** Single-pulse waveform. The vertical axis was calculated from the obtained pulse energy (30.4 µJ) and the pulse waveform. **e** Optical spectrum. **f** $M^2$ (beam quality) measurements (inset: beam profile at the beam waist). **g** Output power stability for 24 hours. The data presented in **c**–**g** were measured at an injection current of 330 mA.

absorber, and each interface and emitting end has a dielectric multi-layer coating. Regarding the VECSEL, a p-DBR (GaAs/AlGaAs), an active layer (InGaAs quantum well), and an n-DBR (GaAs/AlGaAs) were grown on a GaAs substrate with p-/n-contact layers (GaAs) using metal–organic chemical vapor deposition. The reflectance of p-DBR and n-DBR were designed to be 99.9% and 96.1%, respectively. An oxide layer for current and optical confinement was located near the active layer with an aperture of 150 µm diameter created using a selective oxidation process. Electrodes were formed on p-/n-contact layers. To reduce the absorption loss at the wavelength of 940 nm, the GaAs substrate was thinned to 0.1 mm, whose surface was coated with an antireflection coating ($R < 1\%$ at 940 nm). For the mechanically assembled laser, the gain medium was a [111]-cut Yb:YAG with a thickness of 0.5 mm and a doping concentration of 4 at. %. For the integrated chip-scale laser, the thickness was 0.35 mm and the doping concentration was 10 at. %. The saturable absorber was a 0.2 mm thick [100]-cut Cr:YAG with an initial transmittance of 95% for the mechanically assembled laser and 97% for the integrated chip-scale

laser. A short wave-pass filter ($R < 1\%$ at 940 nm, $R > 98\%$ at 1030 nm) and a long wave-pass filter ($R > 98\%$ at 940 nm, $R < 1\%$ at 1030 nm) were coated at one of the two surfaces of the Yb:YAG (the VECSEL side) and at the other surface of the Yb:YAG (the Cr:YAG side), respectively. An OC ($R = 85\%$ and 80% for the mechanically assembled and integrated chip-scale laser, respectively, at 1030 nm) was formed at the laser emitting surface of the Cr:YAG.

## Device fabrication

To fabricate our laser, first, dielectric coating was formed on each surface of the GaAs substrate, the Yb:YAG, and the Cr:YAG using the conventional deposition method[22]. Second, each substrate was bonded one by one to the integrated structure using the wafer bonding technique, whose total thickness was 0.65 mm. After the bonding process, the integrated device was cut into 1 mm square chips by dicing. Finally, the fabricated laser chip was mounted onto a TO-can using Au–Sn solder paste and electrically connected through wire bonding.

**Table 1 | Parameters used for simultaneous rate equations model**

| Symbol | Parameter | Value |
|---|---|---|
| $\eta_i$ | Carrier injection efficiency | 0.99 |
| $I_{off}$ | Offset current | 0.49 A |
| $G_O$ | Initial gain coefficient | $2.6 \times 10^5$ s$^{-1}$ |
| $N_O$ | Carrier transparency number | $1.8 \times 10^5$ |
| $\beta$ | Spontaneous emission coefficient | $6.9 \times 10^{-17}$ |
| $\varepsilon$ | Gain compression coefficient | $9.9 \times 10^{-28}$ |
| $k$ | Scale-factor of the output coupling efficiency | $1.0 \times 10^{-7}$ W |
| $\tau_n$ | Carrier lifetime in the VECSEL cavity | $9.9 \times 10^{-9}$ s |
| $\tau_p$ | Photon lifetime in the VECSEL cavity | $9.8 \times 10^{-12}$ s |
| $I$ | Injected current | 0.91 A |
| $l_{sub}$ | Length of the GaAs substrate | 0.1 mm |
| $t_V = 2(n_{sub}l_{sub} + n_{YAG}l_g)/c$ | VECSEL external cavity round-trip time | $8.4 \times 10^{-12}$ s |
| $A_{Yb}$ | Single-pass absorption rate of Yb:YAG | 18.1% |
| $\sigma_g$ | Stimulated emission cross section of Yb:YAG | $2.2 \times 10^{-20}$ cm$^2$ |
| $\sigma_{SA}$ | Ground-state absorption cross section of Cr:YAG | $4.6 \times 10^{-18}$ cm$^2$ |
| $\sigma_{ESA}$ | Excited-state absorption cross section of Cr:YAG | $8.2 \times 10^{-19}$ cm$^2$ |
| $l_g$ | Length of Yb:YAG | 0.5 mm |
| $l_{SA}$ | Length of Cr:YAG | 0.2 mm |
| $N_{SAi} = -\ln(T_O^2)/(2\sigma_{SA}l_{SA})$ | Total population density of Cr:YAG | $5.6 \times 10^{17}$ cm$^{-3}$ |
| $t_r = 2n_{YAG}(l_g + l_{SA})/c$ | Q-switched laser cavity round-trip time | $8.5 \times 10^{-12}$ s |
| $\tau_g$ | Lifetime of the upper laser level of Yb:YAG | $9.5 \times 10^{-4}$ s |
| $\tau_{SA}$ | Excited-state lifetime of Cr:YAG | $3.4 \times 10^{-6}$ s |
| $\gamma_g$ | Inversion reduction factor of Yb:YAG | 2 |
| $N_i$ | Total population density of Yb:YAG | $5.5 \times 10^{20}$ cm$^{-3}$ |
| $hc/\lambda_p$ | Photon energy | $2.1 \times 10^{-19}$ J |
| $\lambda_p$ | Pumping wavelength | 940 nm |
| $F$ | Intensity adjustment factor | 2.9 |
| $D$ | Diameter of pumping laser | 0.16 mm |
| $V = \pi(D/2)^2 \times l_g$ | Pumping volume | $1.0 \times 10^{-2}$ mm$^3$ |
| $L$ | Nonsaturable round-trip dissipative optical loss | 0.05 |
| $c$ | Vacuum speed of light | $3.0 \times 10^8$ m/s |
| $q$ | Elementary charge | $1.6 \times 10^{-19}$ C |
| $n_{sub}$ | Refractive index of the GaAs substrate | 3.55 |
| $n_{YAG}$ | Refractive index of the laser crystal | 1.82 |
| $R_{OC}$ | Reflectance of the OC | 85% |
| $T_O$ | Initial transmittance of Cr:YAG | 95% |

## Pumping-power absorption model

With experimentally evaluated VECSEL output power ($P_{exp}$), the following equations are derived for the pumping-power absorption ($P_{Yb}$),

$$P_{cav}(R_{eff}) = P_{exp}(R_{eff}) \times \frac{T_{eff} + A_{eff}}{T_{eff}}, \quad (1)$$

$$P_{Yb} = P_{cav}(R_{eff1}) \times \frac{A_{eff1}}{T_{eff1} + A_{eff1}} \cdot \frac{A_{Yb}}{A_{Yb} + A_{GaAs}}, \quad (2)$$

where $P_{cav}$ is the sum of the laser output power and absorbed power in the VECSEL cavity, which is referred to as the potential pumping power. An external cavity can be treated as one effective layer[23], and then, $R_{eff}$, $T_{eff}$, and $A_{eff}$ are introduced as the effective reflectivity, effective transmissivity, and effective absorption rate of the external cavity part, respectively. They are calculated using Fabry-Perot resonance theory[24] under the maximum reflection condition (i.e., antiresonance condition) with parameters $R_2$, $R_{940}$, and $A_{GaAs}$. $P_{cav}$ is obtained for each $R_{940}$ condition in the experiment (Fig. 2b) and is defined as a function of $R_{eff}$ by interpolating the obtained data. As a next step, the pumping-power absorption $P_{Yb}$ is derived. It is assumed that the same power of $P_{cav}$ is obtained at the same effective reflectivity at steady state; therefore, $P_{Yb}$ can be calculated by considering the absorbed portion by the Yb:YAG in the external cavity part when the effective parameters ($R_{eff1}$, $T_{eff1}$, and $A_{eff1}$) are changed according to the external cavity parameters ($R_2$, $R_3$, $A_{GaAs}$, and $A_{Yb}$). In addition to the major external cavity parameters, which are taken into account in this model, further consideration of other factors, such as diffraction and scattering losses, will contribute to realizing more accurate cavity design.

## Simultaneous rate equations model

Table 1 summarizes the parameters used in the simultaneous rate equations model.

The carrier density $N$ and photon density $S$ in the VECSEL cavity of 940 nm are given by

$$\frac{dN}{dt} = \frac{\eta_i(I - I_{off})}{q} - \frac{N}{\tau_n} - \frac{G_0(N - N_0)S}{1 + \varepsilon S}, \quad (3)$$

$$\frac{dS}{dt} = -\frac{S}{\tau_p} - \frac{2\alpha(S, N_g)}{t_V} + \frac{\beta N}{\tau_n} + \frac{G_0(N - N_0)S}{1 + \varepsilon S}, \quad (4)$$

where $\alpha(S, N_g) = A_{Yb} \times S \times (1 - N_g/N_i)$ is the absorbed photon density in the Yb:YAG, which indicates the loss of the VECSEL cavity. Equations (3) and (4) are the modified semiconductor laser rate equations of the VECSEL cavity based on the equations of VCSEL[25]. The parameters listed in Table 1 from $\eta_i$ to $\tau_p$ were extracted from I–L and current–voltage profiles under the maximum output condition shown in Fig. 2b.

The photon density ($\phi$), population inversion density of the Yb:YAG ($N_g$), and ground-state population density of the Cr:YAG ($N_{SA}$) in the passively Q-switched laser cavity of 1030 nm are respectively given by

$$\frac{d\phi}{dt} = \frac{\phi}{t_r}\left[2\sigma_g N_g l_g - 2\sigma_{SA} N_{SA} l_{SA} - 2\sigma_{ESA}(N_{SAi} - N_{SA})l_{SA} - (L - \ln R_{OC})\right], \quad (5)$$

$$\frac{dN_g}{dt} = W(\alpha(S, N_g)) - \frac{N_g}{\tau_g} - \gamma_g \sigma_g c \phi N_g, \quad (6)$$

$$\frac{dN_{SA}}{dt} = \frac{N_{SAi} - N_{SA}}{\tau_{SA}} - \sigma_{SA} c \phi N_{SA}, \quad (7)$$

where $W(\alpha(S, N_g)) = \alpha(S, N_g) \times k \times F / (hc/\lambda_p \times V)$ is the volume pumping rate into the upper laser level, which indicates the pumping intensity of the Yb:YAG. Equations (5)–(7) are the modified rate equations of the passively Q-switched laser cavity based on equations of solid-state lasers[26,27]. The rate equations of the semiconductor and the solid-state lasers can be coupled by the existence of a common $\alpha(S, N_g)$. Therefore, the oscillation mechanism of the proposed laser can be simulated by solving the Eqs.

(3)–(7) numerically. Note that to provide a qualitative description of the behavior of the laser cavities, our model simplifies some conditions of the actual laser configuration. A quantitative analysis will be discussed in future studies.

## Reporting summary

Further information on experimental design is available in the Nature Research Reporting Summary linked to this paper.

## Data availability

The authors declare that the source data generated in this study are provided with this paper.

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

## Acknowledgements

We acknowledge technical support for VECSEL fabrication from Yuta Yoshida, Masato Oishi, Yuta Otoguro, and Naoto Kikuchi at Sony Semiconductor Manufacturing Corporation, for dielectric coating from Yasuro Nakagawa and Shohei Abe at Sony Global Manufacturing and Operations Corporation.

## Author contributions

All authors contributed extensively to the work presented in this paper. M.K. conceived the original concept. M.K., G.H., J.Y., K.T., R.K., and Y.H. designed the basic chip-scale laser structure. G.H. and H.W. designed the VECSEL devices. G.H. fabricated and tested the VECSEL devices. M.S., G.Y., H.T., and Y.I. fabricated the chip-scale laser and performed the measurements. K.T. and J.Y. performed the theoretical analysis and the rate equation simulation. J.Y., K.T., G.H., G.Y., and M.K. wrote the paper. M.K. supervised the project with K.Y.

## Competing interests

M.K., G.Y., G.H., and K.T. are inventors in several patent applications related to this work filed by Sony Group Corporation (including PCT patent application no. PCT/JP2020/043292). The authors declare that they have no other competing interests.
