## [Peer review file · Nature Communications]

REVIEWER COMMENTS

Reviewer #1 (Remarks to the Author):

In the manuscript, a high-peak-power chip-scale semiconductor/solid-state vertically integrated laser with two cavities coupled with each other is presented. The GaAs substrate, the Yb:YAG, and the Cr:YAG were bonded one by one to the integrated structure using the wafer bonding technique. Finally, a compact high-peak-power laser with a peak power of 76.1 kW is realized. So the structure shown here may provide a method to achieve a chip-scale integrated laser. Although it is a combination of several mature technologies that lacks of novelty, I think that it is a difficult and meaningful job. My other comments are listed as follows:

1. How do you get carrier transparency number(N_0) of 1.8×10^5 ? (Page 21)
2. Why is the unit of pump volume(V) mm^{-3} instead of mm^3 ? (Page 22)
3. Please give reasonable explain for the curve oscillation of Fig.2d and Fig.4a.
4. Can the electro-optic conversion efficiency be calculated and how about the thermal effect of VCSEL?
5. How to ensure interface quality and device life by double bonding method? Can the device life test be provided?

Reviewer #2 (Remarks to the Author):

The authors report on a novel concept to realise a fully integrated and miniaturised laser capable of emitting laser pulses with a peak power approaching the 100-kW level.

In my opinion this is a very significant achievement and the concept is novel and original and can potentially be expanded to other material system combinations, i.e. to other emission wavelengths. I am convinced that this source will be useful for a range of application fields and think that the work therefore deserves to be published in Nature Communications.

However, there are a few points I would ask the authors to address in a revised version of the manuscript before publication:

- In general, the language is somewhat hard to follow at times, and I would recommend utilising professional editorial services to improve the overall quality of the paper.

- It is stated that the peak power is estimated from "the measure pulse energy". As this is arguably the most crucial parameter, more details should be given. How exactly was the pulse energy measured?

How was the temporal pulse profile measured, in particular what was the bandwidth of the photodiode and the oscilloscope used? Is it clear that there are no significant pre- or post pulse pedestals present?

- It is stated that "splendid Gaussian profiles are obtained", yet I would argue that a full M^2 measurement should be included. For many applications the beam profile is a crucial parameter.

- More information on the pulse-to-pulse as well as on the long-term stability should be given. I would recommend to include an RF noise spectrum and to perform a proper noise analysis.

- The claim that "it is possible to obtain a shorter pulse width using a semiconductor SA" should be underpinned by simulation results.

Prof. Alex Fuerbach

Macquarie University

Reviewer #3 (Remarks to the Author):

Here, J. L. Yue et al., demonstrate a passive Q-switch laser of a fully integrated pump laser source and gain medium with the chip-size dimension of 1 mm^3 . The novel approach of the laser system comprises two cavities monolithically, a vertically- external-cavity-surface-emitting laser (VECSEL), and Yb:YAG with Cr:YAG. At First, the authors numerically conduct the cavity design of VECSEL and passively Q-switched laser. After that, the authors developed the mechanical assembly laser system and then the fully integrated chip-scale laser. At the available maximum output, the pulse energy is 12.1 uJ with a pulse duration of 384 ps at a wavelength of 1031.1 nm which corresponds to the peak power of 76.1 kW . At a monolithic dimension size of 1 mm^3 , the obtained output power is high extremely.

The presented data are convincing and interesting, and the manuscript is well-organized. I believe that this paper might be recommended for publication in Nature Communications if the authors consider the following comments.

Comments.

1. On lines 232-233, how much is the repetition rate of the laser chip at the output pulse energy of 35.9 uJ ? Would you please explain how to control the repetition rate of the output laser pulse?
2. The threshold current of 370 mA was necessary to start the Q-SW oscillation in the mechanically assembled laser. In contrast, with the fully integrated chip-scale laser, the threshold current was 550 mA . Would you please comment on the reason why the threshold current increases in the fully integrated chip-scale laser?
3. In Fig. 4 and 5, the authors show the cross-section line trace of the beam profile. In addition, the waveform of both axes is modulated slightly. To check the output laser profile, it should be indicated the

far-field pattern of the laser profile image. Would you please add the result of the M square beam profile of the output laser profile?

4. In comparison with the mechanically assembled laser and the fully integrated chip-scale laser, the spectral bandwidth of the output laser pulse in Fig 5. c is broader. What is the reason why the spectral structure is different? Please comment on it. If possible, it is better to add the log scale spectrum.

5. Would you please describe the power scaling with increasing the chip size of this novel scheme in the section of the discussion?

6. Could you give us a comment on the stability of output pulse energy during a couple of hours?

7. Did the authors pay attention to suppressing the parasitic oscillation in Yb: YAG? If the authors have tried any technique to suppress the parasitic oscillation, please describe it.

Dear Reviewers:

Thank you so much for reviewing our manuscript and offering valuable suggestions.

We responded to your comments point by point, and revised the manuscript accordingly. Please confirm the following revision notes and revised manuscript for your further consideration.

Revision Notes

1. Additional data for the chip-scale laser in Fig. 5 have been requested. Unfortunately, the laser chip used in Fig. 5 was damaged and no additional data were available. We fabricated a new laser chip to obtain the additional data, so please let us replace Fig. 5 with data from the new chip. Please also note that the condition of the new laser chip differs from that of the previous laser chip owing to different fabrication preparation conditions. In addition, because the measurement of M^2 was performed in Fig. 5f, the beam profile presented in Fig. 4e is deleted.
2. Please note that all changes in the manuscript text file are highlighted in yellow.

Revision for Reviewer #1:

1. How do you get carrier transparency number(N_0) of 1.8×10^5 ? (Page 21)

As mentioned in **Methods: Simultaneous Rate Equations Model** (Page 20, the carrier transparency number N_0 was derived from the I–L and current–voltage profiles by the method described in Reference 25.

2. Why is the unit of pump volume(V) mm^{-3} instead of mm^3 ? (Page 22)

As you have pointed out, mm^3 is the correct unit, not mm^{-3} . The correction has been made in Table 1 (Page 23) of the revised paper.

3. Please give reasonable explain for the curve oscillation of Fig.2d and Fig.4a.

The oscillations in the I–L curve are currently under investigation. Possible causes could be the mechanical vibration caused by the mechanically assembled configuration, the near-field pattern

change in the VECSEL cavity due to the increased drive current, and the effect of returning light between the boundaries of the laser cavity elements.

4. Can the electro-optic conversion efficiency be calculated and how about the thermal effect of VCSEL?

The electro-optic conversion efficiency of the device can be calculated, but readers can possibly infer the detailed device structure from the data in the paper. As the efficiency data are sensitive information of our company, we respectfully request to keep them private.

5. How to ensure interface quality and device life by double bonding method? Can the device life test be provided?

The bonding interface was made flat to reduce the possibility of optical damages. The output power stability data for 24 hours were obtained and are added to Fig. 5g in the revised manuscript. Longer life tests of the device are under investigation.

Revision for Reviewer #2:

- In general, the language is somewhat hard to follow at times, and I would recommend utilising professional editorial services to improve the overall quality of the paper.

We sent our paper to a professional editorial service for proofreading.

- It is stated that the peak power is estimated from "the measure pulse energy". As this is arguably the most crucial parameter, more details should be given. How exactly was the pulse energy measured? How was the temporal pulse profile measured, in particular what was the bandwidth of the photodiode and the oscilloscope used? Is it clear that there are no significant pre- or post pulse pedestals present?

Details of the pulse energy and temporal pulse profile have been added to Page 13. The pulse energy was measured by a pyroelectric energy meter (Ophir PE9-C). The temporal pulse profile was measured by a high-speed biased fiber-optic detector (Newport Corporation 818-BB-35F) with a bandwidth of 15 GHz and a digital phosphor oscilloscope (Tektronix TDS7404) with a bandwidth of 4 GHz and a sampling rate of 20 GHz. No significant pre- or post-pedestals have been identified at present.

- It is stated that "splendid Gaussian profiles are obtained", yet I would argue that a full M^2 measurement should be included. For many applications the beam profile is a crucial parameter.

The M^2 data were obtained and have been added to Fig. 5f of the revised manuscript. Instead, the beam profile presented in Fig. 4e is deleted.

- More information on the pulse-to-pulse as well as on the long-term stability should be given. I would recommend to include an RF noise spectrum and to perform a proper noise analysis.

The pulse-energy stability data were obtained for two hours and the output-power stability data were acquired for 24 hours. The results have been added to Fig. 5c and g, respectively. We also attempted to obtain an RF noise spectrum, but the signal-to-noise ratio was difficult to evaluate owing to the intrinsic pulse-timing jitters in passively Q-switching mechanisms.

- The claim that "it is possible to obtain a shorter pulse width using a semiconductor SA" should be underpinned by simulation results.

The claim that the pulse width can be shortened using a semiconductor SA deviates from the scope of the present paper. Therefore, we have removed this claim from the **Discussion** section (Page 16) in the revised manuscript.

Revision for Reviewer #3:

1. On lines 232-233, how much is the repetition rate of the laser chip at the output pulse energy of 35.9 μJ ? Would you please explain how to control the repetition rate of the output laser pulse?

As mentioned above, we replaced Fig. 5 with data from the new laser chip. The pulse energy of the new laser chip at 330 mA is 30.4 μJ and the repetition rate is 1.79 kHz. The repetition rate can be controlled by adjusting the injection current value and the solid-state laser cavity design such as the reflectivity of the output coupler and the initial transmittance of the saturable absorber.

2. The threshold current of 370 mA was necessary to start the Q-SW oscillation in the mechanically assembled laser. In contrast, with the fully integrated chip-scale laser, the threshold current was 550 mA. Would you please comment on the reason why the threshold current increases in the fully integrated chip-scale laser?

The I–L characteristics of the new chip-scale laser have been plotted in Fig. 5b. As mentioned in the last paragraph of **Results: Passively Q-switched Laser Demonstration** (Page 14-15), the different oscillation thresholds of the mechanically assembled and chip-scale lasers can be explained by differences in cavity length, the thermal diffusion effect, and the individual characteristics of the devices, in addition to the cavity configuration.

3. In Fig. 4 and 5, the authors show the cross-section line trace of the beam profile. In addition, the waveform of both axes is modulated slightly. To check the output laser profile, it should be indicated the far-field pattern of the laser profile image. Would you please add the result of the M square beam profile of the output laser profile?

As requested, the M^2 data were obtained and have been added to Fig. 5f of the revised manuscript.

4. In comparison with the mechanically assembled laser and the fully integrated chip-scale laser, the spectral bandwidth of the output laser pulse in Fig 5c is broader. What is the reason why the spectral structure is different? Please comment on it. If possible, it is better to add the log scale spectrum.

We have added the log-scale spectra to Figs. 4d and 5e. Why the spectral bandwidths differ between the mechanically assembled and fully integrated chip-scale lasers is currently under investigation. As mentioned in the last paragraph of **Results: Passively Q-switched Laser Demonstration** (Page 14-15), possible causes are differences in the cavity length, thermal diffusion effect, and pumping intensity.

5. Would you please describe the power scaling with increasing the chip size of this novel scheme in the section of the discussion?

We have described the power scaling with increasing chip size on Page 15 of the revised manuscript. If we can consider the effects of injection current distribution and mode matching between the VECSEL cavity and the passively Q-switched laser cavity, then we could increase the chip size for further power improvement.

6. Could you give us a comment on the stability of output pulse energy during a couple of hours?

The pulse-energy stability data for two hours and the output-power stability data for 24 hours were obtained and are added to Fig. 5c and g of the revised manuscript.

7. Did the authors pay attention to suppressing the parasitic oscillation in Yb: YAG? If the authors have

tried any technique to suppress the parasitic oscillation, please describe it.

In the present demonstration, no special technique was used to suppress parasitic oscillations in Yb:YAG.

We would like to thank you again for your valuable time and comments.

Sincerely,

Masanao Kamata, Ph.D.
Tokyo Laboratory 04, R&D Center, Sony Group Corporation
4-14-1 Asahi-cho, Atsugi-shi, Kanagawa, 243-0014 Japan
Phone No: +81-70-7598-6867
Email Address: masa.kamata@sony.com

Dear Reviewers:

Thank you so much for reviewing our manuscript and offering valuable suggestions.

We responded to your comments point by point, and revised the manuscript accordingly. Please confirm the following revision notes and revised manuscript for your further consideration.

Please note that all changes in the manuscript text file are highlighted in yellow.

Revision for Reviewer #1:

1. In the previous version, the proposed laser produces pulses as short as 384 ps, and an estimated peak power of 76.1 kW with a laser chip dimension of 1 mm³. In the current version, the pulse time is longer and the peak power is lower. Were they relevant to the lifetime? Please explain why the stability experiment was performed at the half of peak power.

The relationship between peak power and device lifetime is still under investigation, although the stable operation of several devices with different peak power has been confirmed. In the current device, the laser characteristics, such as pulse width and M^2 , were all measured under the condition with a driving current of 330 mA. Therefore, the stability experiment was also performed under the same condition. Stability and lifetime tests for different driving currents and output peak power are our future research issues.

2. There are 940-nm, 1030-nm as well as 940 nm, 1030 nm, please unify the form. The “-” in some phrases like 1-mm³ could be removed.

All the “-” between the number and the unit have been removed in the revised paper.

Sincerely,

Masanao Kamata, Ph.D.
Tokyo Laboratory 04, R&D Center, Sony Group Corporation
4-14-1 Asahi-cho, Atsugi-shi, Kanagawa, 243-0014 Japan

Phone No: +81-70-7598-6867

Email Address: masa.kamata@sony.com